# Antithymocyte Globulin Inhibits CD8^+^ T Cell Effector Functions via the Paracrine Induction of PDL-1 on Monocytes

**DOI:** 10.3390/cells12030382

**Published:** 2023-01-20

**Authors:** Dragan Copic, Martin Direder, Katharina Klas, Daniel Bormann, Maria Laggner, Hendrik Jan Ankersmit, Michael Mildner

**Affiliations:** 1Department of Thoracic Surgery, Laboratory for Cardiac and Thoracic Diagnosis and Regeneration, Medical University of Vienna, Waehringer Guertel 18-20, 1090 Vienna, Austria; 2Department of Dermatology, Medical University of Vienna, Lazarettgasse 14, 1090 Vienna, Austria

**Keywords:** antithymocyte globulin, inhibitory co-stimulation, single-cell RNA sequencing, T cell activation

## Abstract

Background: Antithymocyte globulins (ATG) are T cell-depleting antibodies used in solid organ transplantation for induction therapy in sensitized patients with a high risk of graft rejection. Previously described effects besides the depletion of T cells have suggested additional modes of action and identified further cellular targets. Methods: We examined the transcriptional changes arising in immune cells from human blood after ex vivo stimulation with ATG at the single-cell level to uncover additional mechanisms by which ATG regulates T cell activity and effector functions. Findings: Analysis of the paracrine factors present in the plasma of ATG-treated whole blood revealed high levels of chemokines and cytokines, including interferon-γ (IFN-γ). Furthermore, we identified an increase in the surface expression of the programmed death ligand 1 (PDL-1) on monocytes mediated by the released paracrine factors. In addition, we showed that this induction is dependent on the activation of JAK/STAT signaling via the binding of IFN-γ to interferon-γ receptor 1 (IFN-γR1). Lastly, we demonstrated that the modulation of the immune regulatory axis of programmed cell death protein 1 (PD1) on activated CD8^+^ T cells with PDL-1 found on monocytes mediated by ATG potently inhibits effector functions including the proliferation and granzyme B release of activated T cells. Interpretation: Together, our findings represent a novel mode of action by which ATG exerts its immunosuppressive effects.

## 1. Research in Context

For individuals with a high risk of graft rejection, induction therapy with cell-depleting antibodies in addition to standard immunosuppressive regimens has been shown to reduce the risk of acute rejection in solid organ transplantation. Antithymocyte globulin (ATG) has previously been shown to deplete T lymphocytes and thus improve graft survival after transplantation. However, recent data suggest additional modes of action which are not restricted to T lymphocytes but also affect myeloid cell populations such as monocytes. The exact mechanisms have not been identified yet. 

## 2. Added Value of This Study

We were able to demonstrate the induction of the immunosuppressive receptor–ligand PDL-1 on monocytes following ex vivo stimulation of human whole blood and purified PBMCs with ATG. Furthermore, we show that the induction is mediated by paracrine actions and activation of the JAK/STAT signaling pathway in monocytes. In addition, PDL 1^+^ monocytes reduce the proliferative rate and secretion of granzyme B of activated CD8^+^ T lymphocytes in vitro. 

## 3. Implications of All the Available Evidence

The identification of this novel immunosuppressive mechanism extends the previous understanding of ATG and its mode of action and offers a novel target for investigation in patient-relevant settings.

## 4. Introduction

The induction and maintenance of immunosuppressive therapy has strongly contributed to reducing graft rejections after solid organ transplantation [1]. In patients with high immunological risk of acute graft rejection, antithymocyte globulin (ATG) is recommended for induction therapy [2,3]. ATG is a mixture of purified polyclonal γ-globulins obtained from the sera of horses or rabbits inoculated with cells from the human thymus [4]. These antibodies bind to a vast array of antigens found on the cell surface of human T lymphocytes as well as other immune cells [5]. ATG depletes T lymphocytes via classical complement-dependent cell cytotoxicity (CDCC), antibody-dependent cell cytotoxicity (ADCC), opsonization of reactive T cells and the induction of the activation-induced cell death pathway (AICD) [6,7,8,9]. Together, these mechanisms are responsible for the rapid and profound lymphopenia observed in patients treated with ATG. However, during or shortly after completion of induction therapy, the number of circulating T cells in the blood gradually increases and reverts to normal values [10]. Interestingly, the recurring T lymphocytes display impaired proliferative capacity even after the termination of ATG therapy [11]. Therefore, additional immunosuppressive mechanisms besides the direct T cell-depleting effects are apparent, including the masking of antigens, increased clearance of T lymphocytes via the reticuloendothelial system and expansion of regulatory T cells (T_regs_) [11,12,13,14]. Moreover, ATG-mediated effects on the immune system are not restricted to direct actions on T lymphocytes, as previous studies demonstrated an induction of B lymphocyte apoptosis [15], immunomodulation of natural killer cells [16] and inhibition of in vitro monocyte-derived dendritic cell maturation [17,18]. However, the underlying mechanisms responsible for the latter effect are not fully understood.

Immunological responses in T cells require the engagement of the T cell receptor (TCR) together with a presented antigen and a second co-stimulatory signal [19]. Depending on the co-stimulatory signal, T cell effector functions are either promoted or dampened [20]. Continuous stimulation of the TCR due to ongoing antigen presentation, as seen after solid organ transplantation, has been shown to upregulate the inhibitory co-receptor programmed cell death protein 1 (PD-1) on the surface of activated T cells [21]. Ligation of this receptor by its ligand programmed death receptor ligand 1 (PDL-1) transduces an inhibitory signal in activated T lymphocytes and decreases their effector functions, including proliferation and cytokine release [22]. Several studies have already reported on the importance of the PD1:PDL-1 axis for the maintenance of autoimmunity and peripheral tolerance after solid organ transplantation [23,24,25]. However, the involvement of the PD1:PDL-1 system in ATG-induced immunosuppression has not been described so far.

In this study, we examined the transcriptional changes in ATG-treated circulating immune cells from human blood at the single-cell level in order to elucidate novel mechanisms by which ATG exerts its immunosuppressive effects. 

## 5. Materials and Methods

### 5.1. Study Design 

The aim of this study was to investigate the changes induced by the ex vivo treatment of human whole blood with ATG and uncover potential novel mechanisms by which ATG regulates T lymphocyte functions. To this end, we used single-cell RNA sequencing to identify transcriptional alterations set in motion by ATG in different immunological subsets of lymphoid and myeloid white blood cells. For scRNA-seq, two donors were sampled per experimental group. 

To verify our bioinformatics data, we performed a series of cell culture- and flow cytometry-based assays. For the in vitro experiments involving purified PBMCs and purified monocytes, the number of individual replicates as well as statistically significant differences are shown in the figure legends, and we included at least three donors per experiment. We examined the effects of ATG-mediated paracrine factors on purified monocytes and identified a crucial role of IFN-γ in the induction of the immune regulatory co-receptor PDL-1. We also delineated the functional contributions of PDL-1^+^ monocytes on activated CD8^+^ T cells in vitro. 

### 5.2. Ethics Statement

This study was conducted in accordance with the Declaration of Helsinki and local regulations. Blood samples were obtained from healthy volunteers who had given their consent to donate prior to participation. The Institutional Review Board of the Medical University of Vienna approved this study (Ethics Committee votes: 1539/2017). All donors provided written informed consent. 

### 5.3. Preparation of Single-Cell Suspension of Human Whole Blood

For scRNA-seq, heparinized human whole blood was drawn from two age-matched male donors. A total of 3 mL of whole blood was either treated with 100 µg/mL ATG (Grafalon; Neovii Biotech GmbH, Gräfelfing, Germany), an equivalent dose of polyclonal rabbit isotype control (ab37415; Abcam, Cambridge, UK) or left untreated. The used dosage corresponded to the blood levels detected in patients undergoing ATG induction [26]. Samples were incubated at 37 °C for 8 h. Next, red blood cells were removed by red blood cell lysis buffer (Abcam). Cells were then washed twice with PBS containing 0.04% bovine serum albumin (BSA; Gibco) and sequentially passed through 100 and 40 µm cell strainers. The LUNA-FL Dual Fluorescence Cell Counter (BioCat, Heidelberg, Germany) and the Acridine Orange/Propidium Iodide Cell Viability Kit (Logos Biosystems, Gyeonggi-do, South Korea) were used to adjust the cell count to 1 × 10^6^ cells/mL with a viability above 90%. 

### 5.4. Gel Bead-in Emulsions (GEMs) and Library Preparation

Single-cell RNA-seq was performed using the 10X Genomics Chromium Single Cell Controller (10X Genomics, Pleasanton, CA, USA) with the Chromium Single Cell 3′ V3 Kit following the manufacturer’s instructions. After quality control, sequencing was performed by the Biomedical Sequencing Core Facility of the Center for Molecular Medicine (Center for Molecular Medicine, Vienna, Austria) on an Illumina HiSeq 3000/4000 (Illumina, San Diego, CA, USA). For donor 1, we detected 2094 cells in total, while for donor 2, altogether 18,257 cells were captured. Raw sequencing data were then processed with the Cell Ranger v3.0.2 software (10X Genomics, Pleasanton, CA, USA) for demultiplexing and alignment to a reference genome (GRCh38). 

### 5.5. Single-Cell RNA Sequencing Data Analysis

Secondary data analysis was performed using R Studio in R (The R Foundation, Vienna, Austria) using the R software package “Seurat” (Seurat v.4.0.0; Satija Lab, New York, NY, USA). Cells were first analyzed for their unique molecular identifiers (UMI) and mitochondrial gene counts to remove unwanted variations in the scRNA-seq data. Cells with feature RNA counts below 100 or above 2500 and more than 10% of mitochondrial genes were excluded from the dataset. Next, we followed the recommended standard workflow for the integration of scRNA-seq datasets [27]. Data were scaled and principal component analysis (PCA) was performed. Statistically significant principal components (PCs) were identified by visual inspection using an elbow plot. Calculation of the Louvain algorithm at a resolution of 0.2 iterations identified a total of 11 communities. The preselected PCs and identified clusters served as the uniform manifold approximation and projection for dimension reduction (UMAP). After the bioinformatics integration of the datasets of untreated, isotype-treated and ATG-treated samples, erythrocytes were removed by excluding all cells with the expression of hemoglobin subunit beta (HBB) > 0.5. Annotation of cell types to the calculated clusters was based on the expression of cell type-specific marker genes that were determined with the “FindAllMarkers” argument in Seurat. We used UMAP plots, dot plots, feature plots, volcano plots and violin plots to visualize differences between the investigated conditions. Normalized count numbers were used to determine DEGs. We applied the “FindMarkers” argument using the default settings to calculate DEGs for clusters of interest between conditions. A log_2_-fold change increase in gene expression above 1 was considered as upregulation, while a decrease below −1 was considered as downregulation. The sets of DEGs were imputed into the Cytoscape [28] plug-in ClueGO [29] to visualize significantly (*p*-value < 0.05, kappa score: 0.4) overrepresented gene ontologies related to immune system processes for the investigated conditions. 

### 5.6. Gene Set Enrichment Analysis (GSEA)

GSEA of gene ontologies was conducted in R using the Bioconductor package “clusterprofiler” [30]. A list of DEGs was calculated comparing every ATG-treated cell type with its matched isotype-treated control. This list was then sorted in descending order for each cell type. For gene annotation, the “org.Hs.eg.db.” package was loaded and executed in R. Non-annotated genes were omitted from further analysis. Using the “gseGO” command, we calculated significantly (*p*-value < 0.05) enriched gene ontologies for biological processes (GO: BP) in the ATG-treated cell types. The default settings for minimal and maximal gene set size as well as the number of permutations were maintained for initial analysis. To adjust the calculated *p*-values and minimize the false discovery rate, we performed a Benjamini–Hochberg correction. Results were visualized by GSEA plots embedded in the “ggplot2” package. 

### 5.7. CellChat Analysis for Ligand–Receptor Interactions

To infer cellular communications between the identified cell types in our scRNA-seq analysis, we used the R package “CellChat” [31]. Firstly, we converted our initial Seurat object into a CellChat object. Next, the CellChat database of receptor and ligand pairs was implemented into the new object. Then, we performed an overrepresentation analysis for genes and possible interactions within the CellChat database. The *p*-value threshold for a significant ligand–receptor interaction was set to <0.05. Cell–cell interactions are displayed as chord diagrams. 

### 5.8. Isolation of PBMCs and Cell Purification

For in vitro assays, PBMCs were isolated using density gradient centrifugation with Ficoll–Paque PLUS (GE Healthcare Bio-Sciences AB, Sweden, Uppsala). Heparinized blood was diluted with phosphate-buffered saline (PBS; Gibco by Life Technologies, Carlsbad, CA, USA) and carefully layered over Ficoll–Paque PLUS. After centrifugation (800× *g*, 15 min, room temperature, with slow acceleration and deceleration), buffy coats containing PBMCs were enriched at the plasma–Ficoll interface. For the purification of CD14^+^ monocytes, CD2^+^ cells and CD8^+^ T cells, we performed magnetic bead separation using CD14, CD2 and CD8 magnetic beads (Miltenyi Biotec, Bergisch Gladbach, Germany) to enrich cells over the QuadroMACS^TM^ Separator (Miltenyi) according to the manufacturer’s protocol. The purity of isolated cells was confirmed by flow cytometry using antibodies against CD14-PE (BioLegend, San Diego, CA, USA; clone: HCD14) and CD8-PeCy5 (BioLegend; clone: SK1) and ranged from 95 to 99%. A complete list of all antibodies used in this study is provided in Appendix A. Acquired cells were counted, diluted to a concentration of 1 × 10^6^ cells/mL and cultured in RMPI 1640 supplemented with 10% fetal bovine serum (FBS; Thermo Fisher Scientific, Waltham, MA, USA) and 1% Pen-Strep (Thermo Fisher Scientific) unless otherwise stated. After 24 h of stimulation with ATG, the PDL-1-FITC (BioLegend, clone: MIH2) antibody was used to assess surface expression on the immune cells.

### 5.9. Proteome Profiler and Enzyme-Linked Immunosorbent Assay

Whole blood samples and isolated PBMCs were treated with ATG (100 µg/mL) or an isotype control (100 µL/mL) and cultured in standard cell culture conditions for 24 h. Next, samples were collected and centrifuged at 1000× *g* for 10 min to obtain cell-free plasma and conditioned media of PBMCs. Supernatants were passed through a 0.22 µm filter before storage at −20 °C until further use. To assess the cytokine profile of the different conditions, plasma- and PBMC-conditioned media from three donors were pooled separately and subjected to the Proteome Profiler Assay Human XL Cytokine Array Kit (R&D Systems, Minneapolis, MN, USA). The assay was performed according to the manufacturer’s instructions. Signals were developed using the Gel Doc XR+ device (Bio-Rad Laboratories, Inc., Hercules, CA, USA) and dot densities for each cytokine duplicate were calculated using the volume tool in ImageLab 6.0.1 (Bio-Rad Laboratories, Inc.). For visualization, the bar plot option of GraphPad Prism was used (v.8.0.1; GraphPad Software, La Jolla, CA, USA) to display differences detected in the ATG-treated plasma and PBMC supernatant as the fold change relative to the controls. In addition, we measured IFN-γ in supernatants of ATG-treated whole blood, purified PBMCs and purified CD8^+^ T cells as well as human granzyme B in co-cultures via ELISA (Bio-techne, R&D, Minneapolis, MN, USA). 

### 5.10. Flow Cytometry

Flow cytometry was performed on a BD FACSCanto II flow cytometer and data were analyzed using FlowJo (10.6.2) software (Tree Star, Ashland, OR, USA). A list of the antibodies used for the detection of cellular epitopes of interest is provided in Appendix A.

### 5.11. INF-γR1 and IFN-γR2 Blockade

Isolated monocytes were cultured at 1 × 10^6^ cell/mL in cell culture medium in the presence of the anti-IFN-γR1 antibody (Bio-techne) for 2 h. Monocytes without added antibody served as controls. Next, the conditioned medium obtained from untreated, isotype-treated and ATG-treated PBMCs was added at a ratio of 1:10. Cells were cultured for a total of 24 h before the flow cytometric assessment of PDL-1 surface expression on monocytes. 

### 5.12. Inhibition of JAK/STAT Signaling in Purified Monocytes

Purified monocytes were treated with 10 µm of the JAK/STAT inhibitor ruxolitinib (MedChemExpress, Princeton, NJ, USA) diluted in DMSO (dimethyl sulfoxide; Merck, Darmstadt, Germany) for 1 h before the addition of conditioned media of untreated, isotype-treated and ATG-treated PBMCs at 1:10. After two hours, one replicate of monocytes was lysed in 1× Laemmli buffer (Bio-Rad Laboratories, Inc.) supplemented with protease and phosphatase inhibitor for the assessment of total Stat 1 (Cell Signaling Technology, Danvers, MA, USA) as well as phosphorylated Stat 1 (Cell Signaling Technology) by Western blot. The remaining cells were cultured for up to 24 h before determining the surface expression of PDL-1 via flow cytometry. 

### 5.13. Western Blotting

Purified monocytes were lysed in 1× Laemmli Buffer (Bio-Rad Laboratories, Inc.) supplemented with protease and phosphatase inhibitor and loaded on 4–15% SDS–PAGE gels (Bio-Rad Laboratories, Inc.). Proteins were transferred on a nitrocellulose membrane (Bio-Rad Laboratories, Inc.), blocked in non-fat milk with 0.1% Tween 20 (Sigma-Aldrich) for 1 h, and incubated with antibodies as indicated in Appendix A at 4 °C overnight. After washing, membranes were incubated with horseradish peroxidase-conjugated secondary antibodies for 1 h at room temperature. Signals were developed with SuperSignal West Dura substrate (Thermo Fisher Scientific) and imaged with a Gel Doc XR+ device (Bio-Rad Laboratories, Inc.). Quantification analysis was performed using the Volume tool in ImageLab 6.0.1 (Bio-Rad Laboratories, Inc.).

### 5.14. Co-Culture of Activated CD8^+^ T Cells with Monocytes

CD8^+^ T cells were isolated from whole blood using CD8^+^ magnetic beads (Miltenyi) and a QuadroMACS^TM^ separator (Miltenyi). In total, 5 × 10^6^ cells/mL were stained with the CellTrace^TM^ violet proliferation kit (Thermo Fisher Scientific) according to the manufacturer’s instructions. Next, anti-CD3 and anti-CD28 T cell activation beads were added to the isolated T cells before placing them in a round-bottom 96-well plate. A total of 5 × 10^5^ T cells was pipetted into each well. After 24 h of activation, the same number of autologous monocytes was added to the T lymphocytes. Prior to addition, monocytes were treated with conditioned media of untreated PBMCs or ATG-treated PBMCs for 24 h and subsequently washed twice. In addition, we incubated PDL-1^+^ monocytes for 1 h with durvalumab (Imfinzi; MedImmune, AstraZeneca, Cambridge, UK), a monoclonal antibody directed against PDL-1, to assess its contribution to the inhibition of CD8^+^ T cell proliferation. Experiments were performed with four independent donors. After 4 days of co-culture, cells were collected and stained for CD8-PECy5 (BioLegend) and assessed for cell proliferation with flow cytometry. Conditioned media of co-cultured cells were collected and stored at −20 °C until further use. 

### 5.15. Statistical Analysis

For single-cell RNA-seq, two donors were analyzed. Negative binomial regression was performed to normalize the data and achieve variance stabilization. The Wilcoxon rank sum test was followed by the Benjamini–Hochberg post hoc test to calculate differentially expressed genes. For in vitro experiments, at least three different donors were used. Data were statistically evaluated using GraphPad Prism v8.0.1 software (GraphPad Software, San Diego, CA, USA). When analyzing three or more groups, ordinary one-way ANOVA and multiple comparison post hoc tests with Dunnett’s correction were performed, with *p*-values < 0.05 considered as statistically significant. Data are presented as the mean ± standard error of the mean (SEM).

## 6. Results 

### 6.1. ATG Alters the Transcriptional Profile of Lymphoid and Myeloid Immune Cells from Human Whole Blood

To elucidate the effects of ATG beyond T lymphocyte depletion, as well as its influences on myeloid immune cells, we investigated the transcriptional changes resulting from treatment with ATG. Single-cell RNA sequencing (scRNA-seq) of untreated, isotype- and ATG-treated (ATG) human whole blood revealed 11 distinct cell clusters in each of the investigated conditions (Figure 1A). Cell types were assigned to clusters based on the expression of established marker genes (Appendix A). We identified different subsets of *CD4^+^* T cells (naive *CD4^+^* T cells, effector *CD4^+^* T cells, Tregs), *CD8^+^* T cells (naive *CD8^+^* T cells, effector *CD8^+^* T cells 1, effector *CD8^+^* T cells 2) as well as NK/T cells, B cells, monocytes, *FCGR3A^+^* monocytes and dendritic cells. The relative cluster frequencies across all conditions were highly similar, with the exception of naive *CD4^+^* T cells, which accounted for 41.81% of all captured cells in ATG as opposed to 26.14% and 28.18% in untreated and isotype-treated cells, respectively (Figure 1A). ATG altered gene expression in all cell types relative to the isotype-treated control (Appendix A). In the lymphoid subsets, we were particularly interested in the influences of ATG on cell types with cytolytic effector functions. Effector *CD8^+^* T cells 1 (104 upregulated, 64 downregulated), effector *CD8^+^* T cells 2 (39 upregulated, 43 downregulated) and NK/T cells (116 upregulated, 105 downregulated) showed considerable changes after treatment with ATG. As they hold high functional resemblance, we next determined the overlapping transcriptional regulations that were induced by ATG in these cell types. Between them, 31 genes were commonly upregulated by ATG (Appendix A), while 29 were downregulated (Appendix A). Amongst the upregulated genes, we identified several members of the chemokine gene family (*CCL2*, *CCL3*, *CCL3L1*, *CCL4*, *CCL8*, *CXCL8*, *XCL1* and *XCL2*), the TNF receptor superfamily (*TNFRSF4* and *TNFRSF9*) and *IFNG* (Appendix A). Interestingly, the myeloid cell types captured in our analysis were also strongly affected by treatment with ATG, resulting in an upregulation of 173, 119 and 76 genes, whereas 97, 77 and 40 were downregulated (Appendix A) in ATG-treated monocytes, *FCGR3A^+^* monocytes and dendritic cells, respectively. Amongst the 33 genes that were commonly upregulated by ATG in these myeloid cell types, we identified the IFN-γ response chemokines *CXCL9*, *CXCL10*, *CXCL11* as well as *CD274* (Appendix A)*,* which encodes for programmed death-ligand 1 (PDL-1) [32]. Monocytes were the most abundant myeloid cell population captured in our analysis and revealed the most transcriptional changes (Figure 1A and Appendix A). Significantly regulated genes with the highest log_2_-fold changes in monocytes treated with ATG compared to those treated with the isotype control are shown in Figure 1B, while volcano plots highlighting the regulations in the other cell types are provided as Appendix A. An overrepresentation analysis for immune system processes affected by genes upregulated in monocytes of ATG-treated whole blood revealed significant associations with cellular responses to interferon-γ, immune cell chemotaxis, antibody-dependent cellular cytotoxicity and the regulation of CD8^+^ T-cell activation (Figure 1C). Genes downregulated in ATG-treated monocytes were associated with immune system processes involved in macrophage activation and migration, amongst others (Figure 1D). In line with the transcriptional alterations, the analysis of the plasma of ATG-treated whole blood showed an increased protein release of several cytokines and chemokines. In total, we detected 12 factors that were significantly increased in the plasma of ATG-treated whole blood compared to isotype-treated controls (Figure 1E), including members of the CCL or CXCL family, IL-2, TNF-α, GM-CSF and IFN-γ. Similar findings were observed in conditioned media (CM) of purified PBMCs treated with ATG, where the increase in CCLs, CXCLs and other cytokines, including IFN-γ, was even more pronounced (Appendix A). 

### 6.2. ATG Induces IFN-γ Production and Release in CD8^+^ Effector T Cell Subsets

As “cellular response to interferon-gamma” was the strongest regulated GO term and several of the identified upregulated cyto- and chemokines were IFN-γ-inducible, we next investigated the regulation and action of IFN-γ after ATG stimulation in more detail. First, we corroborated our findings from the proteome profiler with an IFN-γ ELISA and found that absolute plasma levels were significantly increased after treatment with ATG (575.5 ± 235.1 pg/mL; *p*-value = 0.0005) compared to the isotype and untreated control (both below the assays’ detection limit, Figure 1F). Protein levels of IFN-γ in conditioned media of isolated PBMCs were also significantly higher in ATG-treated cells than in the controls (Untreated: 22.76 ± 22.76 pg/mL; Isotype: below assay detection limit; ATG: 575.0 ± 101.3 pg/mL, *p*-value = 0.0008) (Appendix A). Only the subsets of effector *CD8^+^* T cells as well as the NK/T cells showed increased an expression of interferon-gamma mRNA (*IFNG*) following treatment with ATG (Figure 1G). Next, we aimed to further corroborate the IFN-γ modulating effects of ATG on CD8^+^ T cells. Conditioned media of ATG-treated CD8^+^ T cells revealed significantly higher levels for IFN-γ (3059 ± 173 pg/mL; *p*-value < 0.0001) as compared to the isotype (122 ± 47 pg/mL) and untreated controls (98 ± 28 pg/mL) after purification and stimulation (Figure 1H). Moreover, measurement of IFN-γ in CD8^+^ T cell lysate immediately after isolation (218 ± 80 pg/mL) as well as 24 h after stimulation (untreated 24 h: 326 ± 35 pg/mL; isotype 24 h: 367 ± 56 pg/mL; ATG 24 h: 301 ± 45 pg/mL) showed no upregulation of IFN-γ levels by ATG, suggesting that newly produced IFN-γ is immediately released by the cells (Figure 1H). Together, our data indicate that ATG induces the production of IFN-γ in cytotoxic CD8^+^ T cells, which in turn affects monocyte function.

### 6.3. Surface Expression of PDL-1 Is Increased on Monocytes of ATG-Treated Whole Blood

PDL-1, encoded by CD274, is known for its involvement in the regulation of T cell activity via the engagement of PD-1 [33]. In addition to several chemokines and cytokines, *CD274* was also significantly upregulated by ATG in monocytes, *FCGR3A^+^* monocytes and dendritic cells from whole blood (Figure 2A). Treatment with ATG resulted in a more than 2.5-log_2_-fold upregulation of CD274 expression in monocytes (Figure 2A). Almost 60% of all monocytes showed high levels of CD274 after ATG stimulation (0.32 ± 0.32% in untreated, 1.96 ± 1.613% in isotype-treated and 58.74 ± 10.01% in ATG-treated monocytes) (Figure 2B). These changes were further affirmed by the assessment of surface protein expression of PDL-1 in monocytes treated with 100 µg/mL ATG (Figure 3C). Twenty-four hours after stimulation of whole blood with ATG, 89.68 ± 5.2% (*p*-value < 0.0001) of monocytes were positive for PDL-1, while untreated (4.6 ± 0.5%) and isotype (6.6 ± 0.52%)-treated samples showed little, if any, PDL-1 expression (Figure 2C). Similarly, 88.9 ± 2.35% (*p*-value: < 0.0001) of monocytes in ATG-treated purified PBMCs were positive for PDL-1, as opposed to 2.105 ± 0.46% in untreated and 4.932 ± 0.76% in isotype control-treated PSMCs, respectively (Figure 2C). This effect was not observed at lower dosages of ATG (Appendix A). Together, these data suggest that ATG potently upregulates the expression of *CD274* on monocytes, which translates to an increase in the surface expression of PDL-1 on monocytes from whole blood and purified PBMCs.

### 6.4. IFN-γ Modulates Surface Expression of PDL-1 on Purified Monocytes

To identify the mechanism by which ATG increases PDL-1 expression in monocytes, we first investigated whether ATG-induced PDL-1 expression is a result of a direct or indirect stimulation of monocytes. For that reason, we purified monocytes from whole blood and stimulated them with ATG. The purity of isolated monocytes was assessed by flow cytometry and was above 95% (Appendix A). Interestingly, direct stimulation of purified CD14^+^ monocytes with ATG did not lead to an upregulation of PDL-1 expression (Figure 3A,B). Since our transcriptional data identified the strong regulation of several cytokines, we next investigated whether the modulation of PDL-1 was dependent on soluble factors induced by ATG. We therefore stimulated PBMCs with ATG for 24 h and added the resulting supernatant on purified monocytes. Indeed, monocytes incubated with conditioned media of ATG-treated PBMCs were strongly positive for PDL-1, while monocytes treated with conditioned media of untreated PBMCs showed almost no PDL-1 (Figure 3C,D; untreated PBMC SN: 4.44 ± 0.85%; isotype PBMC CM: 4.12 ± 0.78%; ATG PBMC CM: 89.9 ± 2.07%; *p*-value < 0.0001; for gating strategy, see Appendix A). Furthermore, we evaluated the long-term prevalence of PDL-1^+^ monocytes induced by the conditioned medium of ATG-treated PBMCs. Significantly higher mean percentages of PDL-1^+^ monocytes were detected following treatment with ATG CM compared to the controls for up to 120 h (Appendix A). To investigate the signaling pathway(s) responsible for the observed effect, we next performed a gene set enrichment analysis of the genes regulated in monocytes by ATG. This analysis showed a significant enrichment of genes regulated by interferon-γ (Figure 3E and Appendix A) and corroborated the top two immune system processes detected in our overrepresentation analysis (Figure 1C). In addition, we observed a significant regulation of the IFN-γ signaling pathway network between the effector *CD8^+^* T cell subsets and myeloid cell types, as shown by the chord diagram (Figure 3F), suggesting ligand–receptor communication based on the interactions of *IFNG* with its receptors *IFNGR1* and *IFNGR2*. To validate our bioinformatics data, we investigated the expression and function of the IFN-γ receptors on monocytes. Flow cytometry analysis confirmed the expression of IFN-γR1 and IFN-γR2 on purified monocytes (Appendix A). The addition of IFN-γR1- but not IFN-γR2-blocking antibodies almost completely abolished PDL-1^+^ upregulation in monocytes after treatment with PBMC-conditioned media (95.4 ± 1.58% vs. 16.07 ± 1.86%; *p*-value < 0.0001) (Figure 3H and Appendix A). As the IFNγ signaling pathway network (Figure 3F) identified a possible interaction of T cells and NK cells with monocytes, we next isolated CD2^+^ from PBMCs and stimulated them, as well as the CD2-depleted PBMC fraction with ATG. The CM of ATG-treated CD2^+^ cells significantly increased the percentage of PDL-1^+^ monocytes when compared to CM- or CD2-depleted PBMCs treated with ATG (89.4 ± 6.4% vs. 0.26 ± 0.2%, *p*-value = 0.002, Appendix A). These findings suggest that the engagement of IFNγR1 by IFN-γ is required to induce PDL-1 on purified monocytes.

### 6.5. Inhibition of STAT1 by Ruxolitinib Abolishes Induction of PDL-1 on Monocytes by Paracrine Factors 

Since IFN-γR1 signaling is known to involve the activation of the JAK/STAT pathway [34], we next investigated whether the activation of STAT1 influences PDL-1 expression on monocytes. *STAT1* mRNA expression was upregulated in monocytes, *FCGR3A^+^* monocytes and dendritic cells (Appendix A). Moreover, we detected a significant enrichment of the receptor signaling pathway via JAK/STAT (adjusted *p*-value: 0.0022) in monocytes of ATG-treated whole blood (Figure 4A). Further investigation of the JAK/STAT gene family members involved in the key enrichment of this pathway revealed the strong upregulation of *JAK2*, *STAT1, STAT2 and STAT3* in monocytes (Figure 4B), but also other cell types of ATG-treated whole blood (Appendix A). In contrast to the controls, treatment with conditioned media of ATG-treated PBMCs increased phosphorylation of STAT1 in purified monocytes (Figure 4C). Pre-incubation of purified monocytes with the JAK inhibitor ruxolitinib strongly reduced this effect (Figure 4C). Importantly, the addition of ruxolitinib to monocytes treated with conditioned media of ATG-treated PBMCs strongly inhibited PDL-1^+^ expression (Figure 4D). While 92.23 ± 3.06% of monocytes incubated with ATG PBMC CM were positive for PDL-1, only 4.97 ± 1.84% of monocytes were positive for PDL-1 when cells were pretreated with ruxolitinib (Figure 4E, *p*-value < 0.0001). Taken together, we show that the modulation of PDL-1 surface expression on monocytes by ATG is regulated by IFN-γ and dependent on its binding to IFN-γR1 and downstream activation of the JAK/STAT signaling pathway.

### 6.6. PDL1^+^ Monocytes Inhibit CD8^+^ T Cell Proliferation and Release of Granzyme B In Vitro

Activation of co-stimulatory and co-inhibitory signals influences effector functions of T lymphocytes [35]. The binding of PDL-1 to PD-1 confers an inhibitory signal [36]. Monocytes in the ATG condition displayed a positive enrichment for biological processes associated with the regulation of CD8^+^ T cell activation (adjusted *p*-value: 0.0124), negative regulation of T cell proliferation (adjusted *p*-value: 0.0086) and tolerance induction (adjusted *p*-value: 0.0086) (Figure 5A). *CD274*, *IDO1*, *IRF1*, *HLA-A* and *HLA-E* were identified as core enrichment genes for these processes (Figure 5B). Therefore, we thought to assess whether PDL-1^+^ monocytes are able to suppress effector functions of activated CD8^+^ T cells in vitro. Co-cultures of PDL-1^+^ monocytes with αCD3/αCD28-activated CD8^+^ T cells significantly reduced their proliferative capacity (Figure 5C; for gating strategy, see Appendix A). When incubated with untreated control monocytes, 55.03% ± 2.37% of CD8^+^ T cells divided ≤ 3 times and 44.98% ± 2.37% divided ≥ 4 times after 5 days (Figure 5D,E). In contrast, 81.63% ± 2.98% of activated CD8^+^ T cells divided ≤ 3 times and 18.38 ± 2.98% divided ≥ 4 times (Figure 5D,E; *p*-value = 0.0009) after co-culture with PDL-1^+^ monocytes. This effect was significantly less pronounced when PDL1^+^ monocytes were pretreated with durvalumab, a monoclonal antibody directed against PDL-1, prior to co-culture with activated T lymphocytes (59.78 ± 4.44% displayed ≤ 3 and 40.23 ± 4.44% ≥ 4 divisions (Figure 5D,E; *p*-value = 0.0034). In contrast to CD8^+^ T cells, the addition of CM from ATG-treated PBMCs did not inhibit the proliferation of activated CD8^+^ T cells when co-cultured with PDL1^+^ monocytes (percentage of cells with four or more divisions 74.85 ± 3.71%; *p*-value = 0.0002) (Appendix A). Next, we also measured the protein levels of the effector cytokine granzyme B in supernatants of activated CD8^+^ T cells co-cultured with monocytes. The addition of PDL-1^+^ monocytes resulted in significantly lower levels of granzyme B (7.01 ± 0.58 ng/mL) when compared to the untreated controls (11.41 ng/mL ± 0.79; *p*-value = 0.0046) and anti-PDL-1 antibody pretreated PDL-1^+^ monocytes (10.96 ± 0.26 ng/mL; *p*-value = 0.0077) (Figure 5F). Together, these data show that PDL-1^+^ monocytes inhibit the effector functions of activated CD8^+^ T cells, resulting in reduced T cell proliferation and granzyme B secretion. 

## 7. Discussion 

The immunosuppressive action of ATG is generally attributed to its T cell-depleting properties [7,9,13]. However, increasing evidence from several groups suggests additional modes of action, mediated either directly on distinct T lymphocyte subsets or indirectly via the release of paracrine factors [12,37,38,39]. Using scRNA-seq, we detected numerous alterations in gene expression across lymphoid and myeloid cell types of white blood cells from human whole blood treated with ATG. Notably, classical monocytes displayed significant changes that were tightly connected to responses to IFN-γ. As a result, the surface expression of PDL-1 was increased in these cells, enabling them to functionally impair the proliferative capacity and release of granzyme B of activated CD8^+^ T cells, representing an additional mechanism by which ATG exerts its immunosuppressive actions. 

Here, we provide a broad overview of the transcriptional changes modulated by the ex vivo treatment of human whole blood with ATG, capturing different subsets of lymphoid and myeloid cell types. Transcriptional changes in the lymphoid cells are explainable by the antigen specificity of ATG [40]. Influences on myeloid cell types such as natural killer cells and monocyte-derived dendritic cells have been previously reported by others [17,38]. Dalle et al. showed the increased production of IFN-γ in natural killer cells [38], while Roider et al. reported on the in vitro induction of tolerogenic dendritic cells by ATG [37]. We and others previously showed that PBMCs treated with ATG show an increased production of IFN-γ [41,42]. In line with these reports, our current analysis revealed an upregulation of *IFNG* in NK/T cells after stimulation with ATG. Beyond that, we observed the increased expression of *IFNG* in the subsets of effector *CD8^+^* T cells following treatment with ATG and identified these to be specifically modulated by ATG to increase the production and release of IFN-γ. While IFN-γ is generally considered as a pro-inflammatory cytokine, several reports suggest immunosuppressive roles for IFN-γ [43]. The priming of human mesenchymal stem cells with IFN-γ modulated their immunosuppressive capacity and resulted in clinical improvement and prolonged survival in a murine model of graft vs. host disease [44]. Furthermore, IFN-γ has been shown to confer immunosuppressive actions by modulating the expression of the inhibitory immune receptor ligand PDL-1 on various cell types [45,46,47]. In addition to IFN-γ, numerous other cyto- and chemokines were induced by ATG, including in particular, members of the CCL and CXCL family as well as immunomodulatory cytokines. CXCL9, CXCL10 and CXCL11 have especially been considered as IFN-γ response cytokines and have been reported to modulate the expression of PDL-1 in gastric cancer cells [48]. Their presence might explain why after the inhibition of IFN-γR1, the induction of PDL-1 on monocytes was not completely abolished. Furthermore, these factors bind to the chemokine receptor CXCR3 and transduce their signal via the activation of the JAK/STAT pathways [49], supporting the superior inhibition of PDL-1 induction by the JAK/STAT inhibitor used in our experiments. The overrepresentation of paracrine factors regulated by ATG and the resulting activation of intracellular signaling pathways highlight their relevance in the effects mediated by ATG. Our data further substantiate the importance of paracrine factors released by white blood cells in response to ATG and describe an additional immunosuppressive mechanism beyond the already known depletion of T cells.

The observed increase in the surface expression of PDL-1 on monocytes was mediated by the paracrine signaling of IFN-γ and led to an inhibitory effect on activated CD8^+^ T cells. This mechanism has previously been described as an evasion mechanism of different tumor entities [50,51]. Binding of IFN-γ to IFN-γR1 modulated the expression of PDL-1 on tumor cells and suppressed anti-tumor host immunity, leading to disease progression, increased dissemination and a worse overall outcome [52]. While these regulations are clearly detrimental in oncologic settings, following transplantation, modulation of the PD1:PDL-1 axis can significantly improve graft survival and peripheral tolerance [53]. Pre-clinical work by Borges et al. revealed a significant increase in the graft tolerance of PD-1 overexpressing T cells in a fully MHC-mismatched murine cardiac transplantation model [54]. Interestingly, transplantation of PDL-1 knockout donor hearts into PD-1-overexpressing mice resulted in immediate graft rejection, strongly underlining the importance of the receptor–ligand pair for peripheral tolerance [54]. Moreover, the expression of PDL-1 can be found in other cell types besides myeloid cells. Recently, Bracamonte-Baran et al. reported the increased presence of PDL-1 in graft endothelial cells and their inverse correlation with the infiltration of CD8^+^ T lymphocytes in the myocardial biopsies of heart transplant recipients [55]. Furthermore, they showed that in fully MHC-mismatched mice, a conditional knockout of PDL-1 in graft endothelial cells significantly decreased graft survival when compared to the PDL-1-expressing controls [55]. Our observations are restricted to immune cells from whole blood. Still, ATG and paracrine factors induced by ATG might also affect other cell types found in the vasculature or even in the transplanted organ and similarly modulate PDL-1 or alternative pathways to contribute to graft tolerance by reducing T cell-mediated tissue damage.

The emergence of immune therapeutic drugs for the targeted inhibition of co-stimulatory signals with inhibitory effects on the activation and proliferation of T cells, i.e., PDL-1 and PD-1, amongst others, has advanced the treatment of patients affected by oncologic malignancies [56]. However, severe adverse effects related to excessive activation of the immune system by therapy with immune checkpoint inhibition have been reported in these patients [57]. These immune-related adverse events most commonly involved hepatitis, colitis, myocarditis and dysregulations of endocrine systems including the pancreas and adrenal glands [58]. Interestingly, several reports showed that ATG is able to counteract adverse events caused by immune checkpoint inhibition [59,60,61]. While this effect is in part explainable by its T cell-depleting properties, based on our data, it is tempting to speculate that the induction of PDL-1 on monocytes, or other cell types, potentially contributes to the inhibition of autoreactive T cells, resulting in an amelioration of the reported adverse effects. 

Moreover, transplanted patients are naturally at a higher risk of developing neoplasia due to ongoing maintenance immunosuppressive therapy [62,63]. While there are no large randomized clinical trials available, several case reports described a high risk of graft failure and rejection in solid organ transplant recipients treated with immune checkpoint inhibitors underlining the importance of the PD1:PDL-1 axis for the maintenance of peripheral graft tolerance [64,65]. Whether ATG might prove beneficial in these specific settings, where fine margins between the maintenance of graft survival are confronted with the requirement of anti-tumor immunity, has to be addressed in future studies. Our data show that ATG is able to increase the levels of PDL-1 on monocytes and other myeloid blood cells, and the interaction of these cells with activated T cells results in the reduction in their effector functions, including the proliferation and release of granzyme B. 

Our findings are limited to pre-clinical ex vivo and in vitro data. Studies addressing the role of ATG-induced PDL-1 in myeloid cells and its relevance in patient settings will be needed to clarify to what degree this effect contributes to ATG-mediated graft tolerance. In addition, such studies will be better suited to investigate the influences of repeated administrations of ATG, as performed in standardized induction protocols, and enable a prolonged observation of the described effects. Nevertheless, our data indicate a prolonged positivity of PDL-1 on monocytes treated with a single administration of a conditioned medium of ATG-treated PBMCs. In addition, immunosuppressive induction therapy in patients with high immunological risk of graft rejection usually includes additional drugs such as calcineurin inhibitors, mycophenolate-mofetil and glucocorticoids [66]. Of note, pretreatment of PBMCs with an immunosuppressive dose of hydrocortisone prior to the addition of ATG did not decrease PDL-1 on monocytes (Appendix A). This suggests that the combined use of these agents does not interfere with PDL-1 induction. Further studies are needed to reliably dissect potential interactions between these and other compounds regularly used for immunosuppression in combination with ATG. In conclusion, our study provides a comprehensive overview of the transcriptional changes in immune cells from human whole blood treated with ATG on a single cellular level. We identified an increase in the surface expression of PDL-1 on monocytes as a result of paracrine signaling via IFN-γ/STAT/JAK. Additionally, we showed that PDL-1^+^ monocytes potently inhibit the effector functions of activated CD8^+^ T cells in vitro. Altogether, our data suggest an additional mechanism by which ATG indirectly modulates immunosuppressive actions on cell-mediated rejection in white blood cells. 

## Figures and Tables

**Figure 1 cells-12-00382-f001:**
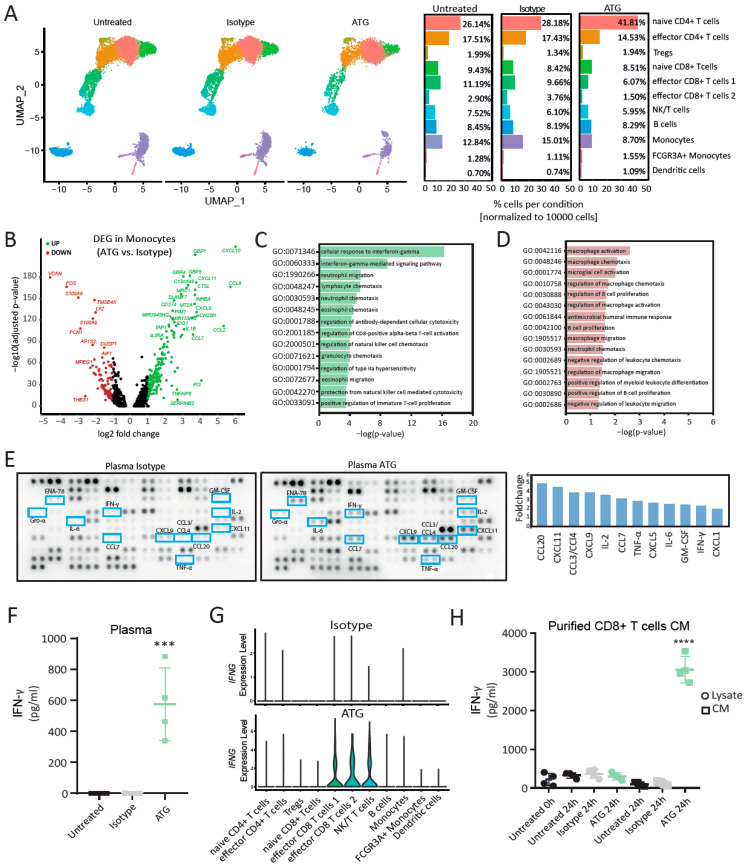
Single-cell RNA sequencing of ATG-treated white blood cells reveals transcriptional alterations in monocytes in response to IFN-γ. (**A**) Uniform manifold approximation and projection (UMAP) plot of untreated (Untreated), isotype-treated (Isotype) and ATG-treated (ATG) immune cells from whole blood reveals 11 distinct cell types present in all conditions. Relative distribution of each cell type across all investigated conditions is shown in the bar plots. N = 2 donors per condition. (**B**) Volcano plot showing differentially up- (green, log_2_FC > 1 and adj. *p*-value < 0.05) and downregulated (red, log_2_FC < −1 and adj. *p*-value < 0.05) genes (DEG) in monocytes (ATG vs. Isotype) with annotations. Adj. *p*-values were obtained after Benjamini–Hochberg correction. Bar plot showing significantly overrepresented immune system processes associated with (**C**) up- and (**D**) downregulated genes in monocytes (ATG vs. Isotype). (**E**) Immunodetection array membrane of plasma of whole blood treated with ATG and the isotype control. Pooled supernatants of 3 donors per condition were analyzed. Proteins with a > twofold increase are shown in blue brackets and highlighted in the bar graph. (**F**) Quantification of IFN-γ by ELISA in plasma from untreated (black), isotype-treated (grey) and ATG-treated (green) whole blood. N = 4 donors. Asterisks denote *p*-values = 0.0005. Ordinary one-way ANOVA was performed followed by Dunnett’s multiple comparisons. (**G**) Expression of *IFNG* across all clusters in the Isotype and ATG. Grade of gene expression is indicated by violin plot height while width represents proportion of positive cells. (**H**) Quantification of IFN-γ by ELISA in lysates (circle) and conditioned medium (square) of untreated (black), isotype-treated (grey) and ATG-treated (green) purified CD8^+^ T cells. N = 4 donors. Asterisks denote *p*-values < 0.0001. Ordinary one-way ANOVA was performed followed by Dunnett’s multiple comparisons.

**Figure 2 cells-12-00382-f002:**
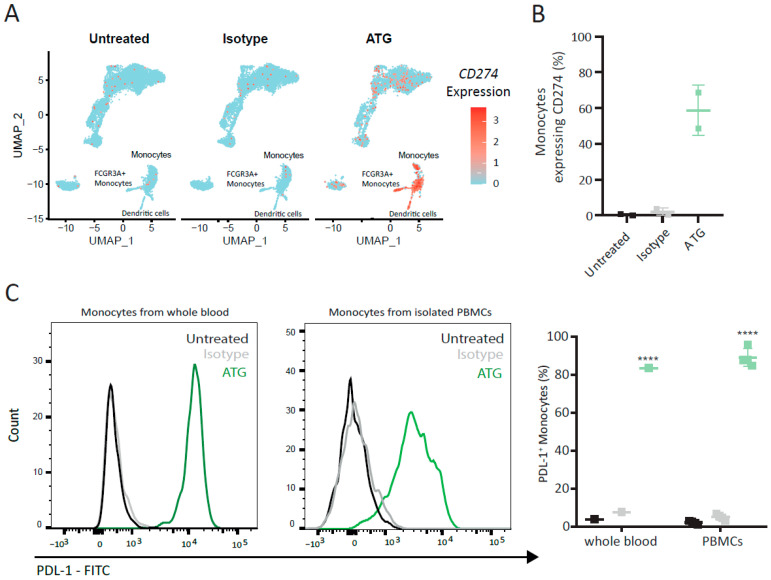
ATG upregulates *CD274* in monocytes from whole blood ex vivo. (**A**) Feature plot for *CD274* of untreated, isotype- and ATG-treated white blood cells. Monocytes, *FCGR3A*+ monocytes and dendritic cells are annotated and the red colour gradient indicates the expression level. (**B**) Bar plot showing the mean percentage of monocytes expressing *CD274* and standard error of the mean (SEM) in Untreated, Isotype and ATG. N = 2 donors. (**C**) Flow cytometry analysis for surface expression of PDL-1 on monocytes from whole blood and purified PBMCs after treatment with ATG. Histogram displaying percentage of PDL-1^+^ monocytes in Untreated (black), Isotype (grey) and ATG (green) from whole blood and purified PBMCs. **** indicates *p*-value < 0.0001.

**Figure 3 cells-12-00382-f003:**
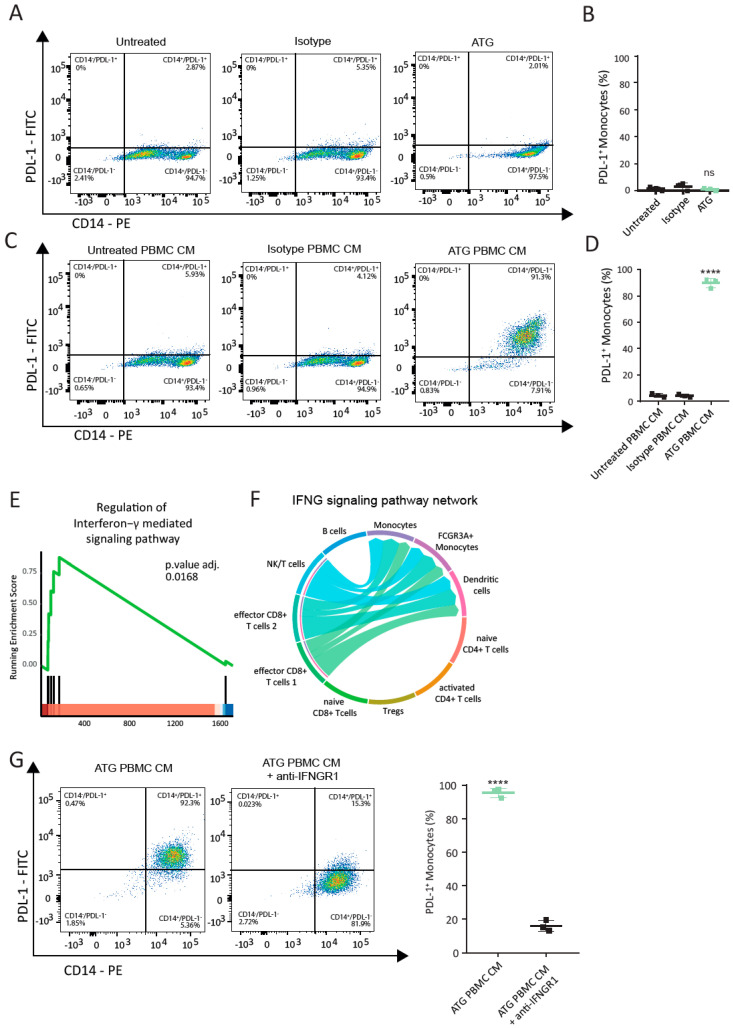
IFN-γ modulates upregulation of PDL-1 on purified monocytes. (**A**) Representative FACS plots of purified monocytes assessed for the surface expression of PDL-1 24 h after treatment with the isotype antibody or ATG. PDL-1^+^ monocytes are displayed in the upper right quadrant. (**B**) Graph displaying the percentage of PDL-1^+^ monocytes after the direct addition of the ATG and controls. (**C**) Dot plots of purified monocytes stained for PDL-1 after treatment with conditioned media of untreated PBMCs, isotype-treated PBMCs and ATG-treated PBMCs and (**D**) graph displaying the percentages of PDL-1^+^ monocytes. (**E**) GSEA enrichment plot for the representative signaling pathway enriched in monocytes from ATG-treated whole blood. The green line depicts the running enrichment score, and the black vertical lines indicate the rank of the genes responsible for the enrichment and position in the list of DEG. (**F**) Chord diagram of *IFNG* signaling pathway network in whole blood cells treated with ATG. (**G**) Representative FACS plots of purified monocytes treated with conditioned media of ATG-treated PBMCs. Surface expression of PDL-1 was assessed with prior inhibition of IFN-γR1 and without. Asterisks denote *p*-values < 0.0001 and were calculated by Student’s *t*-test.

**Figure 4 cells-12-00382-f004:**
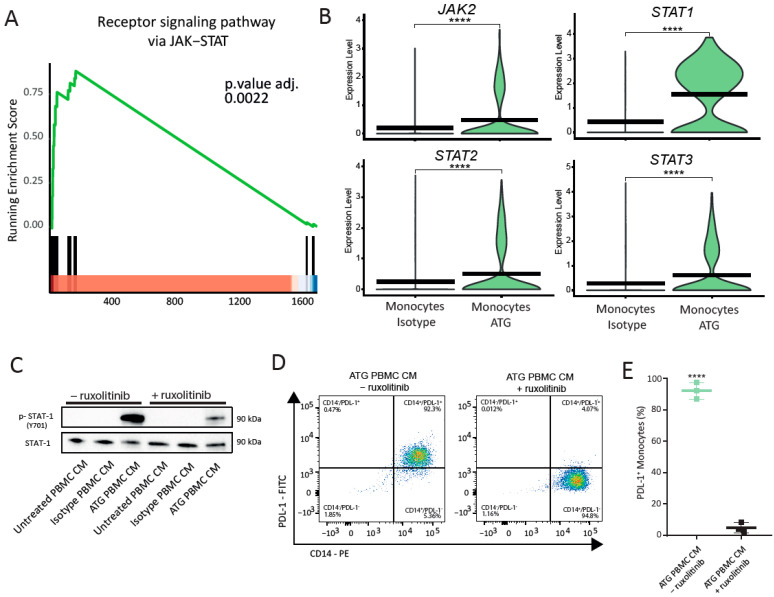
STAT-1 activation is required for the induction of PDL-1 on purified monocytes. (**A**) GSEA enrichment plot for the receptor signaling pathway via JAK-STAT in monocytes from ATG-treated whole blood. The green line depicts the running enrichment score, and the black vertical lines indicate genes responsible for the enrichment and position in the ranked list of DEG. (**B**) Violin plots depicting the differential expression of *JAK2*, *STAT1*, *STAT2* and *STAT3* in monocytes following stimulation with ATG. Expression levels are indicated by violin plot height, while width represents the proportion of positive cells. Crossbars mark mean expression. **** indicates *p*-value < 0.0001. (**C**) Representative Western blots of purified monocytes pretreated with the JAK/STAT inhibitor ruxolitinib for 1 h before stimulation with conditioned media of ATG-treated PBMCs and controls. (**D**) The levels of total STAT1 and phosphorylated STAT1 were assessed after 2 h. (**D**) FACS plots showing the percentage of monocytes positive for PDL-1 24 h after the addition of conditioned medium of ATG-treated PBMCs with and without the prior addition of ruxolitinib. (**E**) Quantitative analysis for surface expression of PDL-1 on monocytes in the investigated conditions for N = 3 donors. Asterisks denote *p*-values < 0.0001. Student’s *t*-test was used to determine significance.

**Figure 5 cells-12-00382-f005:**
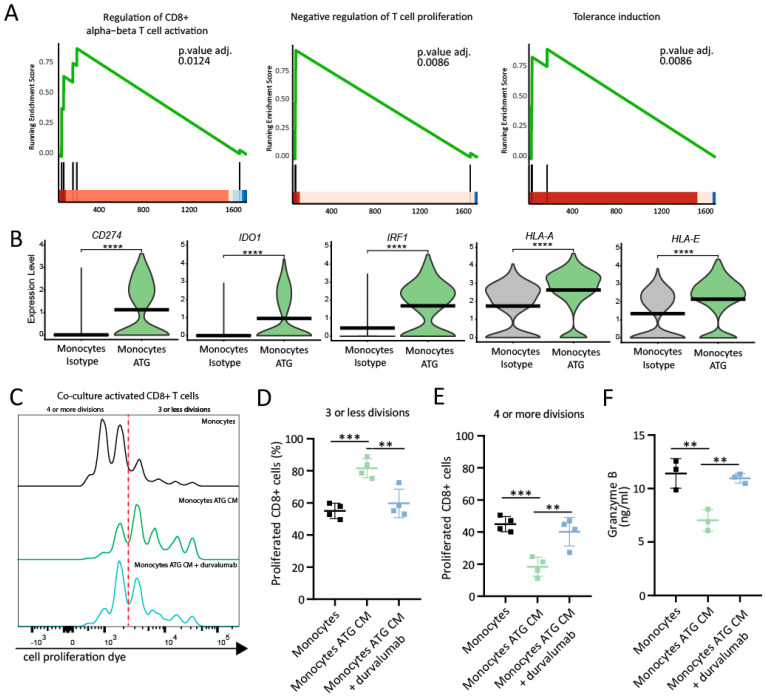
PDL-1^+^ monocytes reduce proliferation and granzyme B release of activated CD8^+^ T cells. (**A**) GSEA enrichment plot for the regulation of T cell functions and tolerance induction in monocytes following treatment with ATG. The green lines depict the running enrichment score, and the black vertical lines indicate the genes responsible for the enrichment and position in the ranked list of DEG. (**B**) Violin plots of core enrichment genes *CD274*, *IDO1*, *IRF1*, *HLA-A* and *HLA-E* in monocytes treated with the isotype control vs. monocytes treated with ATG. Expression levels are indicated by violin plot height, while width represents the proportion of positive cells. Crossbars mark mean expression. **** indicates *p*-value < 0.0001. (**C**) Representative histogram of the proliferative capacity of activated CD8^+^ T cells co-cultured with monocytes (black), PDL-1^+^ monocytes (green) and durvalumab-pretreated PDL1^+^ monocytes (blue). Non-proliferating (furthest right) and proliferating cell populations are reflected by the intensity of cell proliferation dye staining. The dashed red line separates cells with 3 or less divisions from cells with 4 or more divisions. (**D**) Bar graph depicting the mean percentages of CD8^+^ T cells with 3 or fewer proliferations and (**E**) 4 or more proliferations on day 5. N = 4 donors per group. *** *p*-value = 0.0009; ** *p*-value = 0.0034. One-way ANOVA with Dunnett’s multiple comparison was used to determine statistically significant differences. (**F**) Concentration of granzyme B in a conditioned medium of CD8^+^ T cells co-cultured with monocytes assessed by ELISA. ** *p*-value < 0.01.

## Data Availability

Except for the scRNA-seq experiments, all data are available in the main text or in the Appendix A. The scRNA-seq data are accessible upon request.

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
