# Peer review of "Antithymocyte Globulin Inhibits CD8+ T Cell Effector Functions via the Paracrine Induction of PDL-1 on Monocytes"

_cells, 2023, doi:10.3390/cells12030382_

Round 1
Reviewer 1 Report
In this study, The role of ATG in inhibiting CD8 lymphocyte proliferation and producing cytokines such as IFN--γ and paracrine mechanisms with monocytes has been studied.
It was a well-designed study, and the authors presented the results effectively.
Minor revisions are required:
The authors have stated the results of the study in the last paragraph of the Introduction section, but it would be better to state the purpose of the study here.
Please follow the journal's guidelines for writing the manuscript (the western blotting section was written in bold.)
It would be better to display all of the graphs (like Fig.1H and Fig.2 C, etc.).
It is recommended that you avoid duplicating results in text and graphs in the results section.
A minor spelling check is needed (e.g. line 71).
Reviewer 2 Report
Dear authors, it was a pleasure to read about your findings relating to one of the most complicated and important tools in clinical transplantation, anti-thymocyte globulin. Your findings regarding the immunoregulatory potential of the drug are interesting and novel. However, I find that some experiments are lacking in order to fully explore the concept.
Major concerns:
That IFN-g induces PD-L1 and that it can convey a regulatory phenotype is known. More focus should be directed in understanding how rATG mediates these effects. For example it is nicely shown that monocytes exposed to rATG alone do not undertake a PD-L1 positive phenotype, are T cells needed in the PBMC mix in order to induce this change? Other cell types?
A central point in the paper is that rATG is mainly thought to be effective in the T cell compartment as a induction agent. The individual clones of rATG are somewhat know, partly due to the excellent work of Popow et. al (A Comprehensive and Quantitative Analysis of the Major Specificities in Rabbit Antithymocyte Globulin Preparations), and common clones are CD11a/CD18 and CD45 which is broadly expressed on leukocytes. The authors do not comment directly that rATG does indeed bind to monocytes and depletes them from circulation in addition to T cells. Do the authors believe that these clones are significant in mediating the effects seen in the paper? Clones could for example be depleted from rATG to identify which interactions mediate the immunomondulatory effects observed in the work.
It would be interesting to use some of the test conditions setup by the authors to study these more advanced effects, for example is the cell media supernatant generated from rATG stimulated PBMC immuno supressive in general? Or are only derived effects from PD-L1 induction on monocytes? This would also give a more nuanced perspective in some of the assays as a control condition.
Minor:
1. An induction concentration in the blood of rATG is used (100 mikrog/ml). Why not test different concentrations and see if the observed effects are still apparent when a lower concentrations is used closer to that which would be expected in the secondary lymphoid organs (perhaps that is the most important compartment, given that the monocytes are likley to be depleted from circulation?). What dose is needed to induce these changes in monocytes? Is it feasible that this would be happening in secondary lymphoid organs?
2. Why was 5 days chosen in the CD8 proliferation assay? Why is the impact between 3 and 4 proliferation cycles essential? Why not use PI (proliferative index)? When observing the graph the difference seems major between non-treated monocytes and ATG treated monocytes regardless of durvalumab but minor with and without the addition of durvalumab.
3. Low number of replicates generally, some statistics in particular. For example figure 3 G, why is a one-tailed analysis used? For example figure 5 D, perhaps it is not so informative to show to what degree the cells proliferated more than 3 times and less than 3 times in the same graph.
4. Unclear methodology in some instances, were rATG media stimulated monocytes washed prior to addition to the T cells or not? (co-culture of activated CD8+ T cells with monocytes) Or were the CD8+ T cells also exposed to rATG media?
Round 2
Reviewer 2 Report
Dear authors, thank you for the substantial review of your manuscript. The added data supports the primary theories of the work nicely. The argumentation stands stronger after the revision and the work has inspired me to rethink how I view rATG and its effects.
My previous major comments 1-3 are all explored nicely with the new data and I have no further questions.
Comment 5: I think the reasoning for the choice of the duration of the assay is sound and I agree with your assessment. However, I still think that your representative example figure 5C does not reflect your overall dataset in figure 5D/E. If for example PI was used I think that this would be more clear. For the reader it might be informative to nuance that there might be more mechanisms in play that could help regulate the CD8+ T cell response than PD-L1.
Comment 6: Thank you for your comments, I think that the reasoning makes sense. When it comes to figure 5 D/E I still think it is unclear why both graphs are needed, they are mirrors of each other?
I will support the publication of the manuscript in present form.